# Evasion of antiviral bacterial immunity by phage tRNAs

Aa Haeruman Azam[1], Kohei Kondo[2], Kotaro Chihara [1], Tomohiro Nakamura[1], Shinjiro Ojima[1], Wenhan Nie[1], Azumi Tamura[1], Wakana Yamashita[1], Yo Sugawara [2], Motoyuki Sugai [2], Longzhu Cui [3], Yoshimasa Takahashi [1], Koichi Watashi [1] & Kotaro Kiga [1,3] ✉

Retrons are bacterial genetic elements that encode a reverse transcriptase and, in combination with toxic effector proteins, can serve as antiphage defense systems. However, the mechanisms of action of most retron effectors, and how phages evade retrons, are not well understood. Here, we show that some phages can evade retrons and other defense systems by producing specific tRNAs. We find that expression of retron-Eco7 effector proteins (PtuA and PtuB) leads to degradation of tRNA$^{Tyr}$ and abortive infection. The genomes of T5 phages that evade retron-Eco7 include a tRNA-rich region, including a highly expressed tRNA$^{Tyr}$ gene, which confers protection against retron-Eco7. Furthermore, we show that other phages (T1, T7) can use a similar strategy, expressing a tRNA$^{Lys}$, to counteract a tRNA anticodon defense system (PrrC170).

Bacteria evade phage infections by utilizing an immune mechanism referred to as defense system. The retron defense system is composed of reverse transcriptase (RT), non-coding RNA, msrmsd, and accessory protein or an RT-fused domain with various enzymatic functions[1–3]. RT produces satellite msDNA molecules using msd RNA as the template[4]. There are 13 different types of retrons based on their genetic structure and accessory proteins[5]. The accessory protein, which shows large diversity across different retrons[5], is the effector protein that acts to abort phage infection through the inactivation of bacterial growth[1]. However, the defense mechanisms of many retrons remain unknown. Retron-Eco1 from group IIA is the only one among retrons for which the defense mechanism has been characterized thus far[6].

Retron Ec78 (Retron-Eco7), a member of group I-A retrons, utilizes effector proteins derived from the Septu[7] defense system, namely PtuA and PtuB (collectively referred to as PtuAB)[1,3]. Our research demonstrates that Retron-Eco7 employs its effector protein to degrade bacterial tRNA$^{Tyr}$, which is markedly distinct from the previously characterized PtuAB from Septu, known for degrading DNA[8]. Discovered in the 1950s[9], transfer RNAs (tRNAs)

are essential in the central dogma of molecular biology[10]. By the 1960s, tRNAs were identified in bacteriophages[11], and are now known to be prevalent, particularly among virulent phages[12]. Using T5 phages, we showed that T5-like phages utilize their own tRNA$^{Tyr}$ to rescue themselves from retron-Eco7. Further analysis using an anticodon nuclease PrrC170 cloned from a clinical isolate also showed that phage tRNA can be used to rescue phage from PrrC170. In response to antiphage defenses, phages have developed various counter strategies, one of which is to encode proteins that inactivate the host defense[13–16]. Our study highlights the importance of phage tRNA in establishing a counterstrategy for the phage to escape the defense system.

## Results

### Phage genes that inhibit retron function

We previously isolated and characterized a broad host range *Escherichia coli* phage ΦSP15[17], which has a high level of similarity with T5j phage, a wildtype T5 from the phage collection of Jichi Medical University. Using different strains of T5-like phages that infect *E. coli*, we observed a significant genome deletion, particularly

[1]Research Center for Drug and Vaccine Development, National Institute of Infectious Diseases, Shinjuku, Tokyo, Japan. [2]Antimicrobial Resistance Research Center, National Institute of Infectious Diseases, Higashi Murayama, Tokyo, Japan. [3]Division of Bacteriology, Department of Infection and Immunity, School of Medicine, Jichi Medical University, Shimotsuke-shi, Tochigi, Japan. ✉e-mail: k-kiga@niid.go.jp

in the genome of T5 obtained from NBRC (T5n) and the spontaneous mutant of SP15 (SP15m)[17]. Each mutant was shown to carry an approximately 8-kb deletion in their genome, which was later found to encode multiple tRNAs and thus denoted as a tRNA-rich region (TRR) (Fig. 1a). Further bioinformatic analysis demonstrated the prevalence of the TRR in T5-like phages (Supplementary Fig. 1, Supplementary Data 1). We evaluated the ability of the phages to infect bacterial strains that contained different types of antiphage defense systems[3]. In comparison with their respective wild-type strains (T5j and SP15), both deletion mutants, T5n and ΦSP15m, exhibited a notable decrease in infectivity against bacteria carrying retron-Eco2 (Ec67) and retron-Eco7 (Ec78) (Fig. 1b, Supplementary Fig. 2). The TRR of ΦSP15 was further divided into nine fragments (Fig. 1c), each

separately cloned into plasmid carrying the pBAD inducible promoter[18]. They were subsequently transformed into *E. coli* DH10B cells expressing retron-Eco2 or retron-Eco7 (Fig. 1c). The results revealed that fragment 8 (F8 TRR) could rescue T5n and ΦSP15m from both retrons, whereas fragment 6 (F6 TRR) could only rescue phages from retron-Eco7 (Fig. 1d–f, Supplementary Fig. 3a–d). ORF75 of ΦSP15 was discovered to be the genetic determinant within the F8 TRR that enabled the phage to evade the Eco2 and Eco7 retrons (Fig. 1g, h, Supplementary Fig. 4a–d). Meanwhile, tRNA^Tyr in F6 TRR was responsible for phage rescue from Eco7 (Fig. 1h, Supplementary Fig. 4e, f). The function of ORF75 will be characterized in future research. Hereafter we focus on the study of the tRNA^Tyr encoded in F6 TRR.

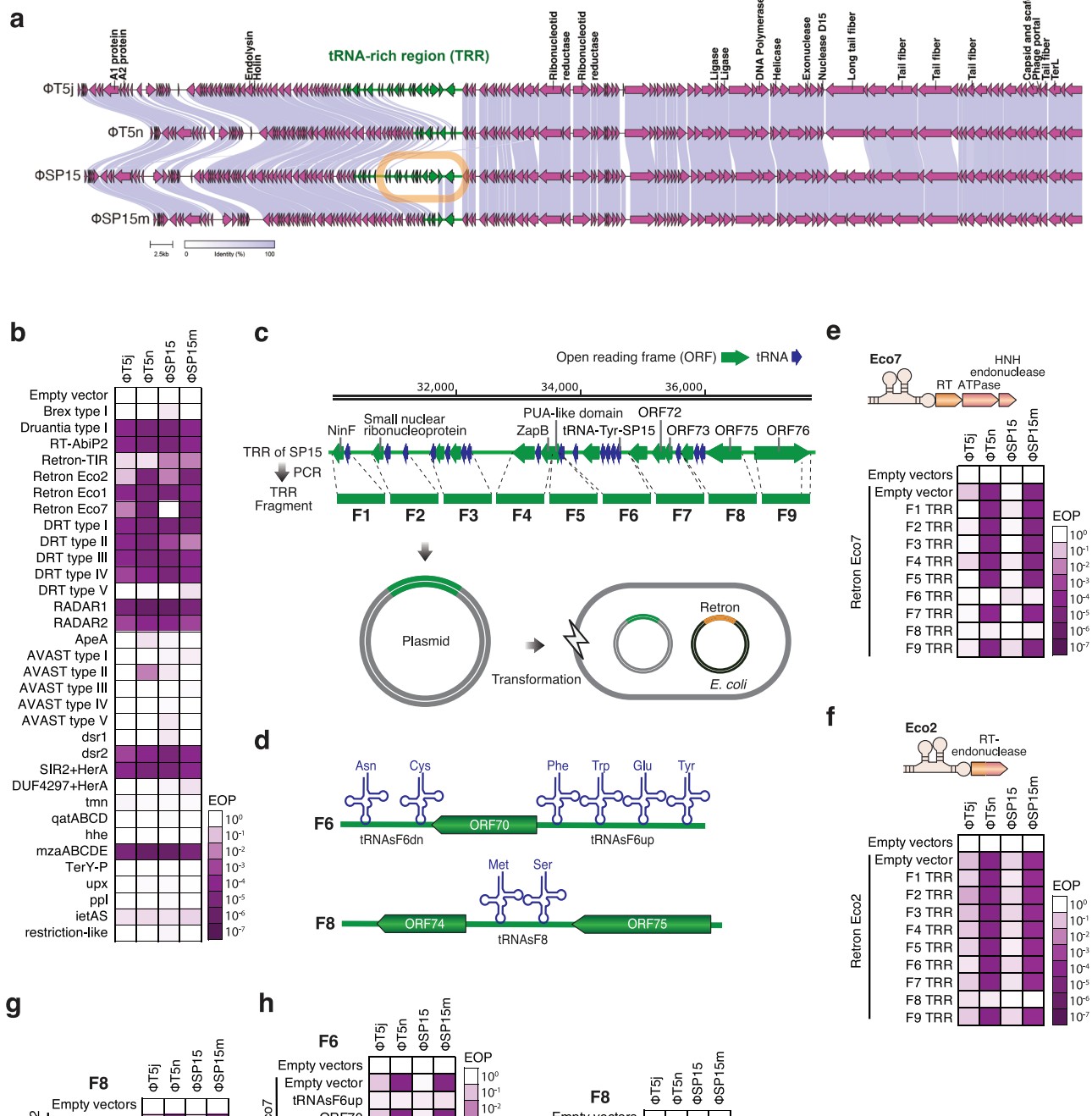

**Fig. 1 | Identification of phage genes involved in retron evasion. a** Genomic comparison of T5 and T5-like phage SP15. The tRNA-rich region (TRR; genomic region of ~8 kb) in T5j and SP15, was missing in T5n and SP15m. Visualized using Clinker[41], the TRR is marked in green, and the highlighted TRR in SP15 is outlined with a brown box. This boxed line indicates the specific area chosen for further experiments. **b** Heatmap depicting the change in the efficiency of plating (EOP) of the phage assay on the bacteria carrying the defense system from Gao et al.[3]. Bacteria carrying pLG001, labeled as Empty vector, served as the negative control. EOP was calculated by dividing the number of phage plaques on bacteria carrying the defense system by the number of phage plaques on bacteria carrying the empty vector. The names of retrons, specifically Ec67 (Eco2), Ec78 (Eco7), and Ec86 (Eco1), were updated based on the retron classification and nomenclature introduced by Mestre et al.[5]. The photograph of the spot assay and the phage count graph are provided in Supplementary Fig. 2a, b. **c** Fragmentation of TRR from SP15. A TRR map, comprising the open reading frame (ORF) indicated in dark green and the tRNA in dark blue, is presented. The numerical annotations above the map correspond to the genomic positions in SP15 (accession number: AP019559). Synthetic fragments of TRR were produced through PCR, assembled into plasmid under pBAD inducible promoter, and subsequently introduced into bacteria that harbor retrons. **d** Genetic organization of the TRR fragment 6 (F6) and 8 (F8). Heatmap illustrating the EOP of phages on bacteria carrying retron-Eco7 (**e**) or Eco2 (**f**) and various TRR fragments; the photograph of the spot assay and the phage count graph are provided in Supplementary Fig. 3a–d. **g** Heatmap illustrating the EOP of phages on bacteria carrying retron-Eco2 and fragmented F8; the photograph of the spot assay and the phage count graph are provided in Supplementary Fig. 4a, b. **h** Heatmap illustrating the EOP of phages on bacteria carrying retron-Eco7 and fragmented F8 and F6. The photograph of the spot assay and the phage count graph are presented in Supplementary Fig. 4c–f. The phage assay used to calculate the EOP presented in this figure was performed in triplicate. Source data are provided as a Source Data File.

## PtuAB of retron I-A exhibits variations in the target molecule

As the inhibition of retron by overexpressing phage tRNA[Tyr] from F6 TRR was specific to retron-Eco7 (Fig. 1e, f), we focused on retron-Eco7 to investigate whether our study could contribute to elucidating the defense mechanism of this retron. Retron-Eco7 belongs to type I-A and has two effector proteins, PtuA, which contains an ATPase domain, and PtuB, an HNH endonuclease[1,3]. We expressed the effector proteins individually (PtuA or PtuB) or together (PtuAB) under the inducible promoter pBAD. PtuAB, but not the singly expressed effectors, was shown to trigger bacterial growth arrest, indicating that PtuA and PtuB are toxins of retron-Eco7 (Fig. 2a, b). To assess the specific component within the retron complex capable of neutralizing the PtuAB toxin, individual retron elements (msdmsr, RT, or PtuAB) were cloned into two different plasmids and co-expressed. Following expression of RT and PtuAB on separate plasmids, RT alone was inadequate at neutralizing PtuAB. Both msrmsd and RT were required for effective neutralization of PtuAB (Fig. 2c, Supplementary Fig. 5). This finding suggests the presence of a tripartite toxin–antitoxin system in retron-Eco7, resembling the previously documented retron-Sen2[2].

The RNA hybridization assay showed that the bacterial tRNA[Tyr] was significantly depleted by retron-Eco7 PtuAB overexpression or during infection of phage (Fig. 2d, e, Supplementary Figs. 6a, 7 and 8). Next, tRNA sequencing confirmed that bacterial tRNA[Tyr], tRNA[TyrU] (tRNA-Tyr-GTA-2-2), and tRNA[TyrV] (tRNA-Tyr-GTA-1-1) were downregulated in bacteria expressing PtuAB (Fig. 2f, Supplementary Fig. 6b–d, Supplementary Data 2). Taken together, these results reveal that retron-Eco7 exerts its protective effect by aborting phage infection through the depletion of bacterial tRNA[Tyr] via PtuAB. However, future research is needed to clarify whether PtuA and PtuB are directly or indirectly responsible for tRNA cleavage.

To assess the universality of this mode of action, we used additional PtuAB variants from the same subfamily, type I-A, retron-Eco4 (Ec83) (Supplementary Fig. 9a–c). In contrast to retron-Eco7, retron-Eco4 demonstrated a broad-spectrum defense activity, providing bacterial protection against a variety of phages (Supplementary Fig. 9d, e). Subsequently, we cloned the PtuAB of Eco4 and assessed its toxicity using the pBAD inducible promoter. Similar to Eco7, the overexpression of PtuAB from retron-Eco4 resulted in bacterial growth arrest (Supplementary Fig. 9f). However, tRNA sequencing data revealed no downregulation of tRNA, suggesting that PtuAB from retron-Eco4 does not target tRNA molecules (Supplementary Fig. 9g, Supplementary Data 3). Although Eco7 and Eco4 belong to the same retron subfamily, these findings indicate that their mechanisms of action differ.

## Phage uses strong promoter to overexpress tRNA[Tyr] and counteract retron-Eco7

As phage-derived tRNA[Tyr] (ΦtRNA-Tyr_SP15) in F6 TRR can rescue phages from retron-Eco7 (Fig. 1h), we hypothesized that ΦtRNA-Tyr_SP15

neutralizes retron-Eco7. As changing the anticodon sequence of ΦtRNA-Tyr_SP15 or mutating the stem-loop sequence of ΦtRNA-Tyr_SP15 abolished the neutralization effect of ΦtRNA-Tyr_SP15 (Fig. 3a–e, Supplementary Fig. 10a, b), we presumed that the function of ΦtRNA-Tyr_SP15 in protein synthesis is essential for retron defense evasion.

Complementation of the exogenous tRNA[Tyr] by T5 tRNA[Tyr] (ΦtRNA-Tyr_T5), *Klebsiella* phage KpP_HS106 tRNA[Tyr] (ΦtRNA-Tyr_KpP_HS106), or *E. coli* DH10B tRNA[Tyr] (Ec_tRNA-TyrU or Ec_tRNA-TyrV) in trans under the SP15-derived tRNA promoter (ΦtRNA-Tyr promoter) successfully restored phage infection to that of ΦtRNA-Tyr_SP15. By contrast, only partial recovery was observed following complementation with the *E. coli* tRNA promoter (Ec_tRNA-Tyr promoter) (Fig. 3f, Supplementary Fig. 10c, d), suggesting that the recovery of phage infection is promoter-dependent. Through the insertion of the red fluorescence protein (RFP) gene under the ΦtRNA-Tyr_SP15 promoter, we observed that this promoter displayed significantly enhanced strength in comparison to RPF expression under either Ec_tRNA-TyrU or Ec_tRNA-TyrV promoters (Supplementary Fig. 11).

## The supplementation of tRNA represents a strategy employed by phages to evade different host defense systems

To investigate whether the utilization of phage tRNA is a widespread strategy employed by phages to evade antiphage defense systems, we extended our analysis to include the PrrC defense system[19]. The PrrC system was screened from the genomes of various carbapenem-resistant bacteria[20] using Defense Finder[21]. We selected a candidate, designated PrrC170, which was isolated from NIID carbapenem-resistant *K. quasipneumoniae* isolate number 170. The toxin gene of PrrC170 harbors an ABC-ATPase domain and has 35% amino acid identity with a well-known PrrC toxin, EcoPrrC (accession number: EER4567187)[22] (Supplementary Fig. 12). We confirmed that PrrC170 is an active antiphage defense system against at least two phages, T1 and T7 (Fig. 3g, Supplementary Fig. 13a, b). We verified that overexpression of the PrrC toxin under the pBAD inducible promoter led to bacterial growth arrest (Fig. 3h). Subsequently, tRNA sequencing was conducted on bacteria harboring the respective plasmid, ultimately elucidating that PrrC specifically targets tRNA[Lys] (Fig. 3i, Supplementary Data 4). Complementation with phage-derived tRNA[Lys] (ΦtRNA-Lys_SP15) resulted in a restoration of phage infection of bacteria carrying PrrC170, whereas complementation with *E. coli*-derived tRNA[Lys] (Ec_tRNA-Lys) showed low restoration of phage infection (Fig. 3i, Supplementary Fig. 13c). An additional complementation experiment using phage-derived tRNA[Lys] from various genera demonstrated that these tRNA molecules enable phages to circumvent the PrrC170 system (Supplementary Fig. 14a–c). Collectively, these findings suggest that the supplementation of tRNA in phages represents a strategy employed by phages to evade host defense systems.

In summary, our experiments with retron-Eco7 and T5-like phages have provided mechanistic model regarding the retron-Eco7 which

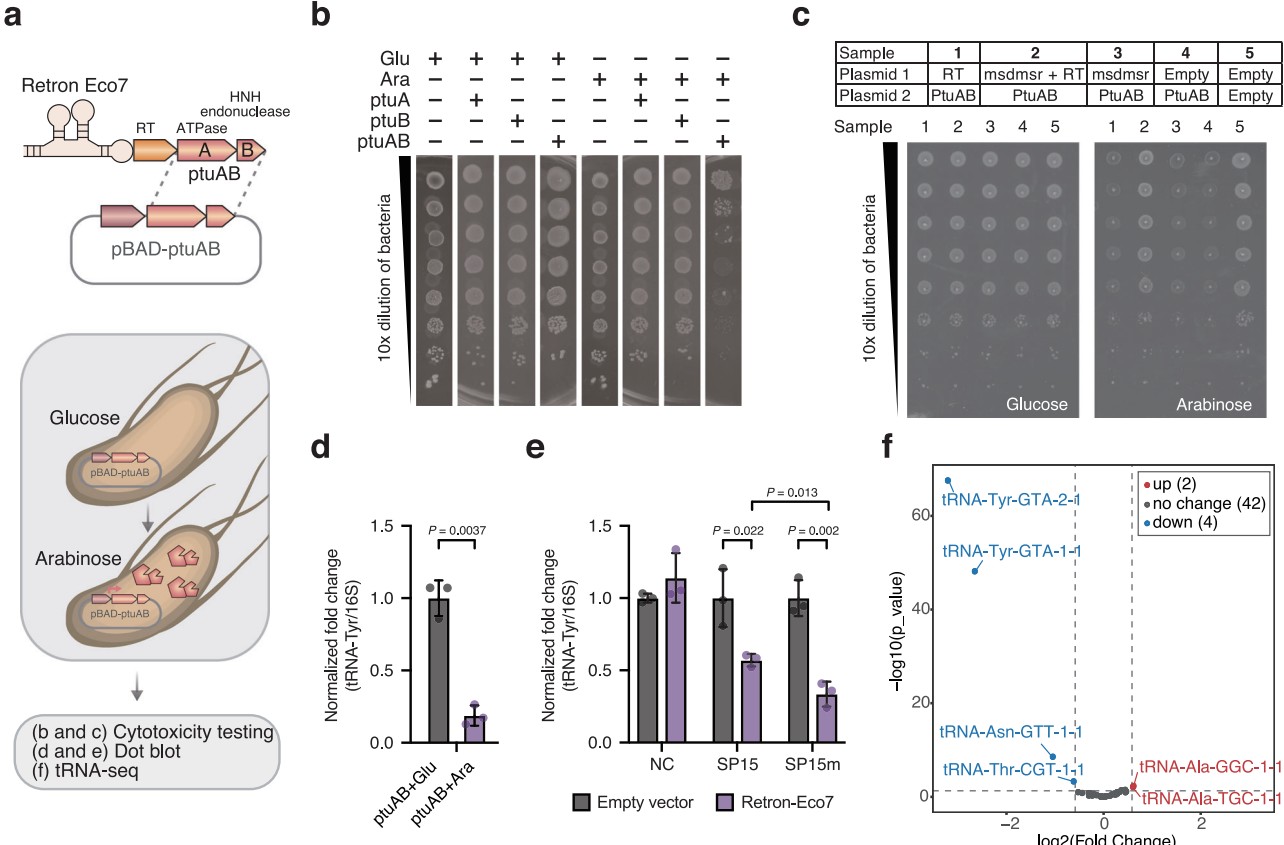

**Fig. 2 | tRNA^Tyr is the cellular target of retron Eco7 effector protein, PtuAB.**
**a** Simplified depiction of the method used to evaluate the cellular targets of PtuAB of Eco7. Effector proteins were expressed under the pBAD inducible promoter. The induced cells were evaluated for their cytotoxicity (**b**), and the reduction in tRNA^Tyr expression using dot blot RNA hybridization (**d**, **e**) and tRNA sequencing (**f**).
**b** Bacterial growth arrest observed in bacteria overexpressing PtuAB. Single expression of either PtuA or PtuB did not promote growth arrest. To induce or repress the expression of PtuA, PtuB, or PtuAB, 0.2% arabinose (Ara) and 0.2% glucose (Glu) were added, respectively. **c** Identification of the antitoxin component in retron-Eco7 via co-expression of a plasmid encoding toxin, PtuAB, and plasmid encoding its antitoxin candidate. The toxin component, PtuAB, was expressed under the pBAD inducible plasmid. The antitoxin candidate (msrmsd, RT, or msrmsd-RT) was constitutively expressed under its native promoter from retron-Eco7. **d** Dot blot RNA hybridization depicting tRNA^Tyr expression levels showed a significant decrease in bacteria where PtuAB was expressed (purple bar) compared to when PtuAB was repressed (dark grey bar). **e** Dot blot RNA hybridization depicting tRNA^Tyr expression levels showed a significant decrease in bacteria infected with phage SP15 or SP15m in the presence of retron-Eco7 (purple bar) compared to those with the empty vector (dark grey bar). The 16S rRNA was used as the control. The dot blot figures, including the negative control using 16S rRNA sense oligonucleotide, are provided in Supplementary Figs. 6 and 7. The

normalized fold change values in (**d**) and (**e**) represent the expression of tRNA relative to 16S rRNA, calculated using ImageJ[36]. These values were normalized by dividing each fold change by the average fold change observed in non-induced PtuAB (**d**) or bacteria expressing an empty vector infected with phage (**e**). The experiments in (**e**) and (**d**) were performed in three biological replicates. Data are presented as mean values ± SD. Statistical significance is indicated by the P-value in the graph. Statistical analysis was performed using a two-tailed Student's t-test, assuming equal variances. Source data are provided in the Source Data file.
**f** Volcano plot depicting tRNA sequencing of bacteria carrying PtuAB under the pBAD inducible plasmid. Two tRNA^Tyr (tRNA-GTA-1 and tRNA-GTA-2) were significantly downregulated in bacteria where PtuAB was induced compared to when PtuAB was repressed. The fold change was calculated based on the total tRNA expression level in bacteria under induction (arabinose added) versus repression (glucose added). The experiment was conducted in two biological replicates. Additional tRNA sequencing data comparing the induced PtuAB to the induced empty vector is available in Supplementary Fig. 6b. The log2(Fold Change) represents the difference in means between two groups, calculated as PtuAB-induce_CPM (Count per million) minus PtuAB-repress_CPM. The statistical significance was determined using the p-value from the exact test based on a negative binomial distribution. No adjustments for multiple comparisons were made.

employs effector protein PtuAB (Fig. 4). Retron-Eco7 functions as a tripartite toxin-antitoxin system, where PtuAB act as toxin genes and retron msDNA and RT protein act as antitoxins. Upon phage infection, retron-Eco7 may be triggered, leading to the release of PtuAB, which causes growth arrest by degrading bacterial tRNA^Tyr before the phage can complete its replication cycle. However, T5 phages likely produce an abundance of their own tRNA^Tyr using a strong promoter, effectively bypassing the degradation of bacterial tRNA^Tyr and ensuring successful phage replication. Further analysis has demonstrated that this tRNA supplementation mechanism is not exclusive to retron-Eco7, thereby broadening our understanding of the capability of phage tRNA to counteract antiphage defenses.

We observed the co-localization of multiple tRNAs in the TRR region. Given that defense systems typically co-reside with other antiphage systems within a single genomic island[3,7,13,23], we sought to determine if other known anti-defense proteins may also be present in this region. However, our bioinformatics analysis revealed that TRR did not display many known anti-defense genes (Supplementary Data 5).

## Discussion

Our investigation into the TRR led to the identification of the cellular target of retron-Eco7 effector proteins, PtuA and PtuB (collectively known as PtuAB), which is tRNA^Tyr. However, retron-Eco4, from the

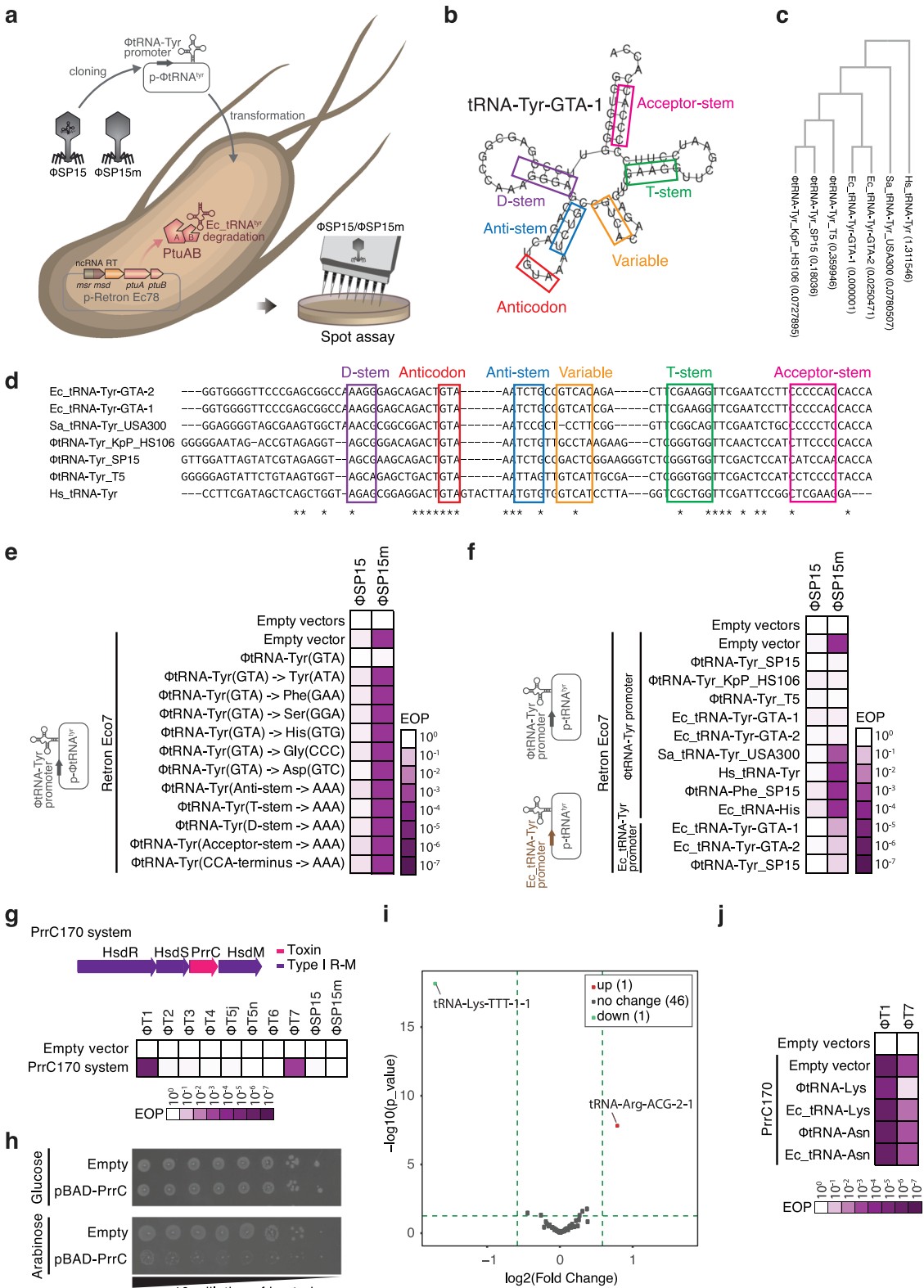

same subfamily, likely does not target tRNA. Notably, PtuAB is also present in Septu defense system[3,7,8,24]. Unlike PtuAB of retron-Eco7, PtuAB of Septu from *E. coli* ATCC25922 was reported to target nicked DNA[8], particularly the genomic DNA of phages, resulting in the termination of phage production. Taken together, PtuAB likely exhibits variation in the target molecule.

tRNAs are known to be present in the genomes of bacteriophages infecting various bacterial genera[12]; however, their precise function

remains elusive. Several hypotheses have been proposed, and the most well-known is codon compensation, in which codons rarely used by the host but required by the phage are supplemented by phage-encoded tRNAs. This hypothesis is supported by the observation that phage-derived tRNAs tend to correspond with codons that are frequently used by phage-encoded genes[12,25]. Recent studies hint at another potential function of phage-derived tRNAs; they were discovered to be used by phages to counteract the depletion of host tRNAs that occurs

**Fig. 3 | Supplementation of phage-derived tRNA is strategy employed by phage to evade defense systems. a** Simplified depiction of the method used to evaluate tRNA complementation on bacteria carrying retron-Eco7. The complementation of tRNA was performed in trans by expressing the tRNA under either the phage tRNA^Tyr promoter or the bacterial tRNA^Tyr promoter and introducing it into bacteria carrying retron-Eco7. **b** RNAFold[42]-based structural prediction of tRNA^Tyr from *E. coli* (Ec-tRNA-GTA-1). The predicted secondary structure of the tRNA is highlighted with colored box lines: D-stem (purple), anticodon loop (red), anticodon stem (blue), variable loop (orange), T-stem (green), and acceptor stem (pink). **c** Phylogenetic tree of the tRNA^Tyr used for the complementation experiment. The DNA alignment was performed using ClustalW[43], and the tree was generated using the bootstrap maximum likelihood method. The value inside the brackets indicates the bootstrap score. **d** Sequence alignment of tRNA^Tyr from T5 (ΦtRNA-Tyr_T5), SP15 (ΦtRNA-Tyr_SP15), *Klebsiella* phage KpP_HS106 (ΦtRNA-Tyr_KpP_HS106), *S. aureus* (Sa_tRNA-Tyr_USA300), *Homo sapiens* (Hs_tRNA-Tyr), and *E. coli* tRNA^Tyr (Ec-tRNA_Tyr-GTA-1 or Ec-tRNA_Tyr-GTA-2). Based on the predicted secondary structure of tRNA^Tyr from *E. coli*, the loop, stem, and anticodon sequences are all highlighted using colored boxes. **e** Heatmap of phage EOP illustrating the mutation in different stem loops and changing the anticodon sequence of ΦtRNA-Tyr_SP15, which abolished the tRNA ability to rescue the phage from retron-Eco7. SP15 and SP15m were used in the phage assay. **f** Heatmap of phage EOP illustrating the expression of tRNA^Tyr from different phages (ΦtRNA-Tyr_T5 and ΦtRNA-Tyr_KpP_HS106) or from *E. coli* in rescuing phages from retron-Eco7 in a promoter-dependent manner. The tRNA was expressed under either the phage tRNA^Tyr promoter (ΦtRNA-Tyr promoter) or the *E. coli* tRNA^Tyr promoter (Ec_tRNA-Tyr promoter). Hs_tRNA-Tyr, Sa_tRNA-Tyr_USA300, and E. coli tRNA^His (Ec_tRNA-His) were

unable to rescue phages from retron-Eco7. The photograph of the spot assay and the phage count graph of the heatmaps in (**e**) and (**f**) are presented in Supplementary Fig. 10a–d. Source data are provided as a Source Data File. **g** Heatmap of phage EOP illustrating the antiphage function of the PrrC170 anticodon nuclease (named after isolate number 170 of carbapenem-resistant *Klebsiella quasipneumoniae*) against at least two phages, T1 and T7. The PrrC170 system comprises PrrC and an associated restriction-modification system type I, cloned in pLG001 plasmid[3] under its native promoter. The photograph of the spot assay and the phage count graph are available in Supplementary Fig. 13a, b. Source data are provided as a Source Data File. **h** Growth arrest observed in bacteria expressing the PrrC toxin. The *prrC* gene was cloned under the pBAD inducible plasmid. **i** Volcano plot depicting tRNA sequencing of bacteria carrying the pBAD-PrrC toxin. The fold change was calculated based on the total tRNA expression level in bacteria under induction (arabinose added) versus repression (glucose added). The log2(Fold Change) represents the difference in means between two groups, calculated as PrrC-induce_CPM minus PrrC-repress_CPM. The statistical significance was determined using the *p*-value from the exact test based on a negative binomial distribution. No adjustments for multiple comparisons were made. **j** Heatmap of phage EOP illustrating the complementation of tRNA^Lys from the SP15 (ΦtRNA-Lys) phage in rescuing phages from the PrrC170 defense system. Complementation of *E. coli* tRNA^Lys (Ec_tRNA-Lys), *E. coli* tRNA^Asn (Ec_tRNA-Asn), tRNA^Asn from SP15 (ΦtRNA-Asn) did not rescue phage from PrrC170. The complementation was performed by expressing the tRNA in trans under phage tRNA promoter and introducing it into bacteria carrying PrrC170. The phage count graph is provided in Supplementary Fig. 13c. Source data are provided as a Source Data File.

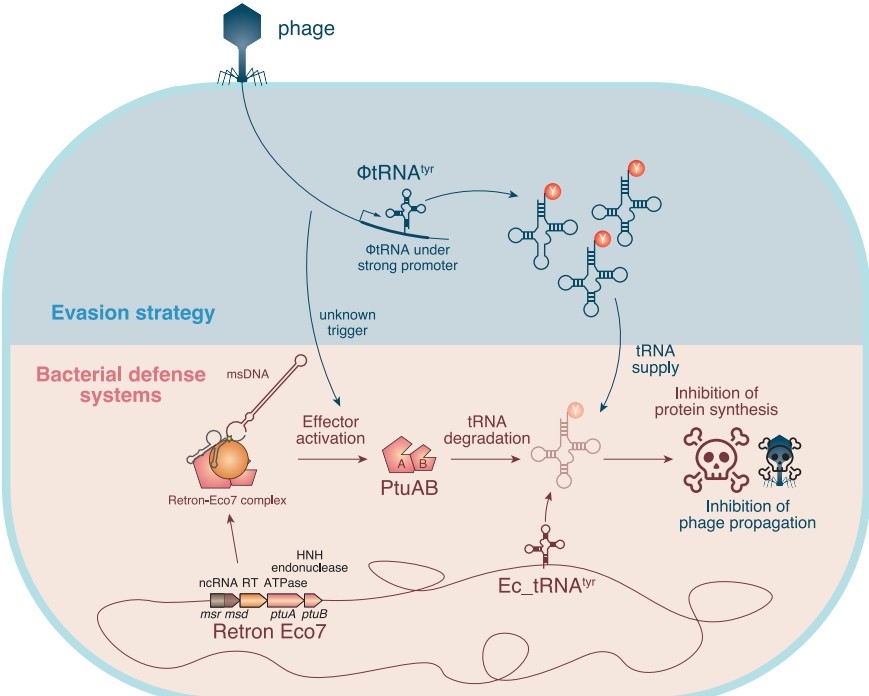

**Fig. 4 | Proposed mechanism of T5 phages circumventing the retron-Eco7 antiphage defense system.** The retron-Eco7 defense system operates as a tripartite toxin-antitoxin complex, where PtuAB acts as the toxin, and retron msDNA along with RT protein serve as the antitoxins. During phage infection, retron-Eco7 may be triggered by an unknown mechanism, potentially leading to the release and activation of PtuAB, which causes bacterial growth arrest by degrading bacterial tRNA^Tyr, thereby preventing the phage from completing its replication cycle. However, T5 phages produce an abundance of their own tRNA^Tyr using a strong promoter, effectively bypassing retron-Eco7 and ensuring successful phage replication.

as a widespread response to phage infection[26,27]. Our data showed that T5 and SP15 phages can no longer infect bacteria protected by the retron-Eco7 defense system when they lose tRNA^Tyr in the TRR, suggesting that the phage-derived tRNA^Tyr is crucial for circumventing the retron-Eco7 defense. Noteworthy, we have identified that the phage utilizes an exceptionally strong promoter to express its tRNA^Tyr, implying that the phage evades retron-Eco7-mediated tRNA^Tyr cleavage by producing a high abundance of tRNA^Tyr. To date, there have been multiple reports on nucleases, such as VapC[28,29], RloC[30], and PrrC[19], which are involved in the toxin–antitoxin system. They are also known to target tRNA and are activated by various stress responses, including phage infection[31].

A recent study utilizing bioinformatic predictions identified mutational patterns in the anticodon loops of tRNAs encoded by mycobacteriophages[32]. These tRNAs are predicted to be resistant to host anticodon tRNases. In line with our discovery of tRNA degradation by Retron Eco7, another group recently demonstrated that phage T5 evades tRNA cleavage by the PARIS immunity through the expression of a tRNA^Lys variant[33]. We believe that while tRNA levels are important, other factors such as the sequence and structure of the tRNA, particularly the loop region, may also play a crucial role in the ability of tRNA^Tyr to rescue phage from retron-Eco7. The *E. coli*-derived tRNA^Tyr (Ec_tRNA-Tyr) and the phage-derived tRNA^Tyr from SP15 (ΦtRNA-Tyr_SP15) differ in both sequence and structure, which could influence how effectively they interact with retron-Eco7. These subtle variations may enhance their ability to suppress retron-Eco7 activity. To confirm this hypothesis, further study is required, and we plan to investigate this in future research.

Additionally, we have demonstrated that phage-derived tRNAs can also bypass the PrrC defense system as well. We noted that the tRNA^Lys used in this study comes from an unrelated phage, which makes the experiment seem artificial. This was due to the limited number of phages we worked with, specifically the T-series phages (T1–T7). PrrC170 targets T1 and T7, but unfortunately, neither of these phages encodes their own tRNA, making it difficult to use them as models. To address this, we decided to use tRNA^Lys from several phages that belong to different genera. Our results show that phage-derived tRNA^Lys from various sources can help phages overcome the PrrC system. Therefore, we believe that in nature, phages may use tRNA as one of their strategies to counteract host defense systems like PrrC.

Taken together, these findings strongly suggest the utilization of phage tRNA to evade defense systems is a widespread strategy employed by phages. Most bacteria encode multiple defense systems, with an average of 5 systems per genome[21,23]. This could be one of the reasons why T5-like phages, which carry multiple types of tRNAs in their genome, exhibit exceptionally broad host range[17].

We noted that the depletion of tRNA^Tyr during phage infection was significantly different between bacteria carrying retron-Eco7 infected with SP15 and those infected with SP15m (Fig. 2e). This difference is likely due to the presence of ORF75 in the wild-type SP15, which may reduce the defense activity of retron-Eco7. However, we are currently unable to determine the detailed mechanism by which ORF75 inhibits retron-Eco7.

# Methods

## Media and buffers
All experiments were conducted using Luria Bertani (LB) broth (10 g polypeptone, 10 g sodium chloride, and 5 g yeast extract per liter) or LB agar (LB supplemented with 1.5% agar). Phosphate-buffered saline (PBS; 8.0 g NaCl, 0.2 g KCl, 1.44 g $Na_2HPO_4$, and 0.24 g $KH_2PO_4$) was used for the dilution of the bacterial solutions, and sodium-magnesium (SM) buffer (5.8 g NaCl, 0.2 g $MgSO_4.7H_2O$, 50 mL 1 M Tris-HCl (pH 7.5), and 5 mL of 2% (w/v) gelatin solution) was used for phage storage and phage dilution.

## Bacterial strains and phages
Bacterial strains and phages used in our experiments are listed in Supplementary Data 6. Defense systems were expressed in *E. coli* DH10B. Bacteria were grown in LB at 37 °C with shaking at 200 rpm unless specified otherwise, and the appropriate antibiotics were added. Chemical competent *E. coli* DH10B cells were prepared according to a previous study[18]. For exogenous genes expressed in *E. coli*, tetracycline (10 μg/mL), ampicillin (100 μg/mL), or chloramphenicol (20 μg/mL) was used to ensure plasmid maintenance. Plasmids used in this study are listed in Supplementary Data 9.

## Isolation and preparation of phage stock
Phage SP15 was previously isolated from sewage influent obtained from a municipal wastewater treatment plant in Tokyo using the double-layer agar plating method[17,34] with *E. coli* O157:H7 as the propagation host. The SP15 mutant (SP15m) was previously isolated from an in vitro co-culture of O157 and SP15 in the presence of fosfomycin[17]. T5n was obtained from the Biological Research Center, National Institute of Technology and Evolution (Tokyo, Japan). T5j was obtained from the Department of Infection and Immunity, Division of Bacteriology, Jichi Medical University (Shimotsukeshi, Japan). Phages were propagated and purified using a previously described method[35]. Briefly, the purified phage was propagated by mixing 1% O157 overnight culture in liquid LB and incubated overnight at 37 °C. Host cells were removed through centrifugation (5000 × g, 10 min, 4 °C) before performing phage concentration using the polyethylene glycol 6000-NaCl method. Finally, the phage solution was filtered through a 0.2-μm filter and used immediately or kept at 4 °C.

## Plasmid and strain constructions
Plasmids carrying 33 different defense systems as well as one empty vector (pLG001–pLG034)[3] were obtained from Dr. Feng Zhang (Broad Institute, Boston, MA, USA). Plasmids encoding retron Ec83 (Eco4) was synthesized and cloned at Genewiz (South Plainfield, NJ, USA). For co-expression of retron and TRR fragment, the retron-Eco7 and Eco4 were cloned under their native promoters using a plasmid with pSC101 origin of replication (ori). The TRR fragments for retron-Eco7 was constructed using pKLC23[18] as a backbone with the pBAD-inducible promoter and pA15 origin of replication (ori). TRR fragments for retron-Eco2 were constructed using pKLC83[18], with pBR322 ori as a backbone, along with the pBAD-inducible promoter. To construct the plasmid carrying different tRNA^Tyr shown in Fig. 3f, the oligonucleotides of the tRNAs were synthesized at Eurofins (Tokyo, Japan) and introduced into the plasmid using NEBuilder® HiFi DNA Assembly (New England Biolabs, Ipswich, MA, USA, catalog number #E2621). The plasmid carrying tRNA^Lys from various phages in Supplementary Fig. 14 was synthesized at Eurofins (Tokyo, Japan) under the pBR322 ori.

To construct a plasmid with the PrrC toxin–antitoxin (TA) system, the presence of the PrrC TA system was searched in the genome of 400 NIID clinical isolates using Defense finder[21]. The PrrC from carbapenem-resistant *K. quasipneumoniae* isolate number 170 (NIID accession number: JBEAAAI-19-0008) was used for further experiments. The predicted PrrC TA system, which consists of four genes (HsdR, HsdS, PrrC, and HsdM), including 300 nucleotides before the start codon of HsdR, was cloned into plasmid pLG001. Plasmid assembly was performed using NEBuilder® Hifi DNA assembly (New England Biolabs, Ipswich, MA, USA, catalog number #E2621). Plasmid constructs were transformed into *E. coli* DH10B using the heat shock method.

To construct a plasmid expressing the RFP under the tRNA promoter, we predicted the tRNA promoters using the BPROM prediction tool (http://www.softberry.com). The predicted promoters from *E. coli*-derived tRNA-Tyr (tRNA_Tyr-GTA-1 and tRNA_Tyr-GTA-2) and phage-derived tRNA-Tyr (ΦtRNA-Tyr_SP15) were amplified from the *E. coli* DH10B genome and the phage SP15 genome, respectively. The *rfp* gene, including the ribosome-binding site and the *rrnB* T1 terminator sequence, was amplified from the pKLC23-RFP plasmid[18]. The plasmid backbone was obtained from pKLC83[18]. The three fragments were PCR amplified using Q5 High-Fidelity 2× Master Mix (New England Biolabs, catalog number #M0492S), and the plasmids were circularly assembled using NEBuilder® HiFi DNA Assembly. The fluorescence intensity of RFP was quantified using the GloMax® Explorer Multimode Microplate Reader (Promega, Madison, USA) and was subsequently normalized with respect to the optical density ($OD_{600}$). The oligo primers, synthetic DNA, and synthetic plasmid used in this study are listed in Supplementary Data 6–9.

## Bacteriophage spot assay

Spot assay was performed using fresh culture that was prepared by inoculating 1% overnight bacterial culture into LB supplemented with appropriate antibiotic(s) until an $OD_{600}$ of ~2 was reached. A bacterial lawn for spot assay was prepared using 100 µL fresh culture in 4 mL LB top agar (LB, 0.5% agarose, 1 mM $CaCl_2$) and poured onto LB plate. Phage was serially diluted using SM buffer in 96-well plates. Next, 3 µL of phage solution was tenfold serially diluted and dropped onto the plate using a multichannel pipet and incubated at room temperature until it dried, followed by incubation at 37 °C overnight. After overnight incubation, formation of lysis zones (plaques) was recorded. Efficiency of Plating of a phage is calculated by comparing the number of plaque-forming units (PFUs) on a test strain to the number of PFUs on a reference strain.

## Cytotoxicity assay for PtuAB and PrrC

PtuAB from retron-Eco7 (pLG008) or PrrC from PrrC170 were cloned into the pKLC23 plasmid under the pBAD-inducible promoter using NEBuilder® Hifi assembly (New England Biolabs). For PtuAB from Eco4 (pNK83), the PtuAB was cloned into the pKLC83a plasmid, a derivative of pKLC83 with the pBAD-inducible promoter. The resulting plasmid construct was introduced into *E. coli* DH10B using the heat shock protocol. A single bacterial colony that carries PtuAB was inoculated in 2 mL LB with chloramphenicol (for pKLC23) or ampicillin (for pKLC83a) and was incubated overnight at 37 °C with shaking at 200 rpm (until $OD_{600}$ was ~5). Overnight cultures were serially diluted eight times (tenfold) in LB. Next, approximately 3 µL of each dilution was spotted on an LB plate containing appropriate antibiotics and either 0.2% arabinose, 0.2% glucose, or no sugars. The plates were incubated at 37 °C overnight.

## TA assay of retron-Eco7

To identify the specific components within the retron complex capable of neutralizing the PtuAB toxin, individual retron components (msrmsd, RT, or PtuAB) were cloned into two different plasmids and co-expressed. The toxin component, PtuAB, was expressed under the pBAD inducible plasmid from pKL83a. The antitoxin candidate (msrmsd, RT, or msrmsd-RT) was constitutively expressed under its native promoter from retron-Eco7 (from plasmid pLG008). A plasmid carrying msrmsd was created by deleting RT and PtuAB from pLG008, while plasmid carrying RT was created by deleting msrmsd and PtuAB. For cytotoxicity assay, a single bacterial colony that carries PtuAB and antitoxin candidate was inoculated in 2 mL LB with chloramphenicol and ampicillin and was incubated overnight at 37 °C with shaking at 200 rpm (until $OD_{600}$ was ~5). Overnight cultures were serially diluted eight times (tenfold) in LB. Next, approximately 3 µL of each dilution was spotted on an LB plate containing appropriate antibiotics and either 0.2% arabinose, 0.2% glucose, or no sugars. The plates were incubated at 37 °C overnight.

For toxicity assay of bacteria co-expressing the retron-Eco7 component in two inducible plasmids. The toxin component, PtuAB, was expressed under the pBAD inducible promoter, whereas the antitoxin component RT/msrmsd/both RT and msrmsd were expressed under the pATc inducible promoter. The antitoxin candidate was continuously expressed by adding 50 ng/mL of anhydrous tetracycline.

## Dot Blot RNA hybridization

An overnight culture of *E. coli* DH10B harboring plasmid pBAD-PtuAB from Retron-Eco7 was inoculated into 10 mL LB supplemented with chloramphenicol (20 µg/mL) until an $OD_{600}$ of 0.3 was reached. The bacteria were subsequently induced with 0.2% arabinose (to induce PtuAB expression) or 0.2% glucose (to repress PtuAB expression) and cultured for 1 h at 37 °C with shaking at 200 rpm. Following incubation, the culture was centrifuged at $6000 \times g$ for 5 min, and the pellet was washed twice with PBS buffer. Total RNA was extracted using a miRNA isolation kit (Qiagen, Hilden, Germany, Catalog number #217004), according to the manufacturer's recommendation. The total RNA was submitted for dot blot RNA hybridization service of Genostaff Inc. (Tokyo, Japan).

For the RNA dot blot analysis of bacteria infected with phage, bacteria carrying pLG008 (retron-Eco7) or pLG001 (empty vector) were infected with SP15 or SP15m for 20 min. Bacteria without phage infection was used as negative control. The samples were then centrifuged at $10,000 \times g$ for 10 min and washed twice with SM buffer. The bacterial pellet was subjected to RNA extraction using a miRNA isolation kit (Qiagen, Hilden, Germany, Catalog number #217004), according to the manufacturer's instructions. The extracted RNA was then submitted for dot blot RNA hybridization at Genostaff Inc. (Tokyo, Japan).

Purified RNA was denatured at 65 °C for 10 min and twofold serially diluted. Diluted RNAs were spotted onto HyBond N+ membrane (Cytiva, Marlborough, MA, USA, Catalog number #RPN1210B) and the membranes were baked at 80 °C for 1 h. Baked membranes were pre-hybridized with DIG EasyHyb (Roche, Basel, Switzerland, Catalog number #1603558) at 68 °C for 1 h followed by hybridization with 10 pmol/mL of DIG-labeled oligonucleotides complementary to target RNA for 15 h. Membranes were washed with 2× saline sodium citrate (SSC)/0.1% SDS twice at room temperature for 10 min, 0.1× SSC/ 0.1% SDS twice for 15 min, and with TBS-T at room temperature for 5 min. The temperature conditions for hybridization and washing were adjusted according to the probe. Phage-tRNA-tyr-new-AS was set at 50 °C, *E.coli*-tRNA-tyr-AS at 60 °C, and 16S rRNA-AS at 60 °C. The sequences of each probe are as follows: Phage-tRNA-tyr-new-AS: 5'-DIG-AACCACCCGAGACCCTTCCGAGTCGG-3', *E.coli*-tRNA-tyr-AS: 5'-DIG-TCCCTTTGGCCGCTCGGGAACCCCACC-3', and 16S rRNA-AS: 5'-DIG-GATTCCGACTTCATGGAGTCGAGTTGCAGACTCCAATCCG-3'. Thereafter, membranes were blocked with blocking buffer (Roche, Catalog number # 1096176) at room temperature for 30 min and probed with 10,000-diluted anti-digoxigenin-AP, Fab fragments (Roche, Catalog number #1093274) in blocking buffer at room temperature for 1 h. Membranes were washed with TBS-T three times at room temperature for 10 min and rinsed with detection buffer (0.1 M Tris-HCl, 0.1 M NaCl, pH 9.5.) Signals were detected using 100-diluted CSPD substrate (Roche, Catalog number # 1655884.).

The dot blot results were quantified using ImageJ[36] software. The quantified data were then normalized by dividing each value by the average value obtained from the negative control. In Fig. 2d, the negative control was PtuAB without induction (Glucose added). In Fig. 2e, the negative control was bacteria carrying an empty vector and infected with phage. Statistical analysis for Fig. 2d, e was performed using the *t*-test available in GraphPad Prism software.

## tRNA sequencing

tRNA sequencing was performed by Filgen, Inc. (Nagoya, Japan). Sequencing, data curation, and bioinformatic analysis were performed by Arraystar Inc. (Rockvile, MD, USA, project codes #J_120222-tRNA-seq-16 and #J_190923-tRNAseq-16). Total RNA from each sample was quantified using NanoDrop ND-1000 (Marshall Scientific, Hampton, NH, USA). tRNAs were purified from total RNA samples and m1A&m3C demethylated before being partially hydrolyzed according to the Hydro-tRNAseq method. The partially hydrolyzed and re-phosphorylated tRNAs were next converted to small RNA sequencing libraries using NEBNext® Multiplex Small RNA Library Prep Set for Illumina® kit (New England Biolabs). Size selection of ~140–155-bp PCR-amplified fragments (corresponding to ~19–35 nt tRNA fragments size range) was performed. The resultant tRNA-seq libraries were qualified and quantified using the Agilent 2100 BioAnalyzer (Santa Clara, CA, USA). We then equally mixed all libraries and sequenced for 50 cycles on an Illumina NextSeq 500 system using NextSeq 500/550

High-Output v2 kit (75 cycles) following the manufacturer's instructions.

Sequencing quality was examined using the FastQC software, and trimmed reads (passed Illumina quality filter, trimmed 3′-adaptor bases by cutadapt) were aligned to the cytoplasmic mature-tRNA sequences obtained from the GtRNAdb BWA[37] software. For tRNA alignment, the maximum mismatch was 2[38]. The tRNA expression profile was analyzed based on uniquely mapped reads and including mapped reads. The differentially expressed tRNAs were screened based on the count value using the R package edgeR[39]. Principal component analysis, correlation analysis, hierarchical clustering, scatter plots, Venn plots, and volcano plots were performed with the differentially expressed tRNAs in R or Python environment for statistical computing and graphics. The tRNA sequencing raw data and process data for PtuAB from retron Eco7, PtuAB from retron Eco4, and PrrC toxin of PrrC170 are available on the Gene Expression Omnibus repository under accession numbers GSE229290, GSE256077, and GSE256078, respectively.

### TRR extraction from T5-like phages and anti-defense protein search

A total of 250 T5-like phage genomes were collected using online blast. Conserved upstream and downstream sequences of TRR were manually determined and the conserved sequence positions of TRR in each T5-like phages were detected using local blast with an *e*-value threshold <1e$^{-5}$. TRR sequences were extracted using the "subseq" command option of seqkit. Extracted TRRs were annotated using prokka version 1.14.6, and tRNA encoded in TRR were detected using tRNAscan-SE version 2.0.9.

Predicted anti-defense protein sequences in TRR were collected based on previously reported anti-defense proteins[40] and are listed in Supplementary Data 5. Homologous protein sequences were retrieved using online BLAST to enrich the databases. Local blastp was used to find anti-defense candidate proteins with an *e*-value threshold <1e$^{-5}$.

### Reporting summary

Further information on research design is available in the Nature Portfolio Reporting Summary linked to this article.

## Data availability

The raw data for the heatmap and phage counting are presented in the Source Data file provided with this paper. The genome of T5j is available in the NCBI database under accession number AY543070. Genome of T5n is available in the NCBI database under accession number AY692264. The genome of SP15 is available in the NCBI database under accession number NC_048627 (https://www.ncbi.nlm.nih.gov/nuccore/1859677195). The tRNA sequencing raw data and process data for PtuAB from retron Eco7 is available on the Gene Expression Omnibus (GEO) repository under accession numbers GSE229290. The tRNA sequencing raw data and process data for PtuAB from retron Eco4 is available on the GEO repository under accession number GSE256077. The tRNA sequencing raw data and process data for PrrC toxin of PrrC170 are available on the GEO repository under accession number GSE256078. Source data are provided with this paper.

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

## Acknowledgements

We thank Dr. Nishimasu Hiroshi from The University of Tokyo for the fruitful discussion during our manuscript preparation. This work was supported by the Japan Agency for Medical Research and Development (grant No. JP23wm0325065 to A.H.A., K.Kondo, K.C. and K.Kiga, JP21fk0108496, JP21wm0325022, JP22fk0108532, JP24fk0108698 to K.Kiga, JP21gm1610002 to K.Kiga and L.C., JP22fk0108562 and JP23fk0108599 to K.C.) and JSPS KAKENHI (Grant No. 21H02110 and 21K19666 to K.Kiga; 23K13876 to A.H.A; 22K20575 to S.O.; 23K19475 to K.C.). The funders had no role in the study design, data collection and analysis, decision to publish, or preparation of the manuscript.

## Author contributions
A.H.A. provided funding, designed and conducted the experiments, analyzed data, and drafted the manuscript. K.Kondo and T.N. provided expertise in bioinformatic analysis. K.C. and S.O. provided funding, conducted experiments. W.N., A.T and W.Y. conducted experiments and contributed to data collection. Y.S., M.S., L.C., Y.T., and K.W. supervised the study and provided funding. K.Kiga supervised the study, provided funding, and critically reviewed and approved the final manuscript.

## Competing interests
A.H.A., Y.T., K.W., and K.Kiga are co-inventors on a pending patent submitted by the National Institute of Infectious Diseases, which is based on the results reported in this paper. The remaining authors declare no competing interests.
