## [Transparent Peer Review file · Nature Communications]

Evasion of antiviral bacterial immunity by phage tRNAs

Corresponding Author: Dr Kotaro Kiga

This manuscript has been previously reviewed at another journal. This document only contains reviewer comments, rebuttal and decision letters for versions considered at Nature Communications.

Version 0:

Reviewer comments:

Reviewer #1

(Remarks to the Author)

The authors adequately answered our comments.

There should still be a bit of work of some parts of the text to enhance readability.

I29: why "the defense system" and not defense system

I34: why executor and not effector

I106: simplify title

I77: refers to fig 1.G is it not 1.E or 1.F ?

I163: Maybe spell out tRNA rich region instead of TRR

Reviewer #3

(Remarks to the Author)

In the revised manuscript by Kiga and colleagues, the authors have eliminated a section of data focusing on mechanistic characterization of an anti-retron protein, and focused the manuscript on evasion of retron immunity by tRNA expression. In its current form, the manuscript is quite thin and only lightly characterizes a few aspects of the phage-host arms race, leading to few substantiated conclusions. There are repeated discrepancies between the data presented and their interpretation in text. As it stands the work does not meet expected standards in the field. My detailed comments are below.

1. Elimination of Rad data – In a previous version of the manuscript the authors described the discovery and characterization of an anti-retron factor called Rad. The original data led to a relatively confusing model, so the authors opted to eliminate the section on Rad. However, in the current manuscript, Figs. 1c, d, e, f, g and h all still include experimental data relating to the discovery of Rad. These panels all include data related to the inhibition of retron-immunity by TRR fragment F8, which encodes Rad (Fig. 1c even still says Rad in the gene label). But the manuscript text makes no interpretation of the mechanism by which fragment F8 rescues phage infection in the presence of Eco2 and Eco7 retrons. I do not see how the data as presented here can be divorced from the context of the discovery and characterization of Rad.

2. Figure 2 is related to the mechanism of phage interference by the Eco7 retron. The authors find that expression of two retron effector proteins, ptuA and ptuB, are sufficient to stimulate toxicity in the absence of phage. Further, they find that the RT and msmsd components of the retron neutralize this toxicity. Finally, the authors find that either expression of PtuAB or phage infection in the presence of the retron lead to depletion of tRNA-Tyr. Two issues with this dataset are:

a. Minor - The data do not directly demonstrate that Ptua and B are responsible for tRNA cleavage. The effect could be indirect and occur through activation of another enzyme.

b. Major – there is only a very modest change in tRNA content during phage infection by the retron-resistant SP15 vs the retron-sensitive SP15m. How do the authors account for the dramatic difference in retron susceptibility between these two phages? Is there more to it than simply tRNA levels? Do tRNA sequences matter (as they seem to for resistance to PrC in

Fig. 3).

3. The authors observe Eco7 retron resistance upon expression of either phage tRNA-Tyr or host tRNA-Tyr from a plasmid using the phage promoter, suggesting that restoring tRNA levels restores infection. The authors claim that expression of the same tRNAs from a plasmid with the host tRNA promoter has a weak effect. However to my eye the difference is quite modest, my conclusion would be that expression of the tRNA from either promoter provides substantial retron resistance. Are the tRNAs being overexpressed regardless, since they are encoded on plasmids? If conclusions are drawn about tRNA levels and retron resistance, then tRNA levels should be measured under these different rescue conditions.

4. PrrC tests – If I understand correctly, the authors observe resistance to the tRNA-cleaving PrrC system (which protects against T1 and T7 infection) upon expression of a tRNA-Lys from an unrelated phage (SP15). This seems very artificial, and does not provide evidence of a phage that encodes a tRNA to overcome the PrrC system.

Manuscript Title: Virus encode tRNA to evade bacterial immunity
Reviewer Comments & Author Rebuttals

Reviewer #1 (Remarks to the Author):

The authors adequately answered our comments.

Dear Reviewer 1,

Thank you for your feedback and effort to review our manuscript. We have addressed all the required revisions accordingly to improve readability throughout the manuscript.

We appreciate the suggestions and believe these revisions enhance the clarity of the manuscript.

There should still be a bit of work of some parts of the text to enhance readability.

129: why "the defense system" and not defense system

Answer:

We have revised "the defense system" to "defense system" for clarity (Line 29).

134: why executor and not effector

Answer:

We changed "executor" to "effector" to better align with common terminology. (Line 34).

1106: simplify title

Answer:

The title has been simplified for improved readability (Line 111).

177: refers to fig 1.G is it not 1.E or 1.F ?

Answer:

The figure reference has been corrected to Fig. 1E and 1F, as appropriate (Line 78).

1163: Maybe spell out tRNA rich region instead of TRR

Answer:

We have revised line 163 to spell out "tRNA-rich region" instead of using the abbreviation "TRR" for clarity (Line 167).

Reviewer #3 (Remarks to the Author):

In the revised manuscript by Kiga and colleagues, the authors have eliminated a section of data focusing on mechanistic characterization of an anti-retron protein, and focused the manuscript on evasion of retron immunity by tRNA expression. In its current form, the manuscript is quite thin and only lightly characterizes a few aspects of the phage-host arms race, leading to few substantiated conclusions. There are repeated discrepancies between the data presented and their interpretation in text. As it stands the work does not meet expected standards in the field. My detailed comments are below.

Dear Reviewer 3,

Thank you for your thorough and constructive feedback. We greatly appreciate your detailed comments, which have been instrumental in helping us improve the manuscript.

In response to your concerns, we have carefully addressed the issues raised and incorporated some of the answers into the revised manuscript. We believe these revisions significantly enhance the clarity of the work.

1. Elimination of Rad data – In a previous version of the manuscript the authors described the discovery and characterization of an anti-retron factor called Rad. The original data led to a relatively confusing model, so the authors opted to eliminate the section on Rad. However, in the current manuscript, Figs. 1c, d, e, f, g and h all still include experimental data relating to the discovery of Rad. These panels all include data related to the inhibition of retron-immunity by TRR fragment F8, which encodes Rad (Fig. 1c even still says Rad in the gene label). But the manuscript text makes no interpretation of the mechanism by which fragment F8 rescues phage infection in the presence of Eco2 and Eco7 retrons. I do not see how the data as presented here can be divorced from the context of the discovery and characterization of Rad.

Answer:

We have change "Rad" to "ORF75" (Fig 1C), Additionally we have added a sentence "The function of ORF75 will be characterized in future research. Hereafter we focus on the study of the tRNA-Tyr encoded in F6 TRR." (Lines 73-74).

2. Figure 2 is related to the mechanism of phage interference by the Eco7 retron. The authors find that expression of two retron effector proteins, ptuA and ptuB, are sufficient to stimulate toxicity in the absence of phage. Further, they find that the RT and msrmsd components of the retron neutralize this toxicity. Finally, the authors find that either expression of PtuAB or phage infection in the presence of the retron lead to depletion of tRNA-Tyr. Two issues with this dataset are:

a. Minor - The data do not directly demonstrate that PtuA and B are responsible for tRNA cleavage. The effect could be indirect and occur through activation of another enzyme.

Answer:

We have ensured that we do not claim that PtuA and B are directly responsible for tRNA cleavage. We also have added a sentence “Future research is needed to clarify whether PtuA and PtuB are directly or indirectly responsible for tRNA cleavage” in the manuscript (Lines 97-99).

b. Major – there is only a very modest change in tRNA content during phage infection by the retron-resistant SP15 vs the retron-sensitive SP15m. How do the authors account for the dramatic difference in retron susceptibility between these two phages? Is there more to it than simply tRNA levels? Do tRNA sequences matter (as they seem to for resistance to PrrC in Fig. 3).

Answer:

We believe that while tRNA levels are important, other factors such as the sequence and structure of the tRNA, particularly the loop region, may also play a crucial role in the ability of tRNA-Tyr to rescue phage from retron-Eco7. The *E. coli*-derived tRNA-Tyr (Ec_tRNA-Tyr) and the phage-derived tRNA-Tyr from SP15 (Φ tRNA-Tyr_SP15) differ in both sequence and structure, which could influence how effectively they interact with retron-Eco7. These subtle variations may enhance their ability to suppress retron-Eco7 activity. To confirm this hypothesis, further study is required, and we plan to investigate this in future research (Lines 194-201).

3. The authors observe Eco7 retron resistance upon expression of either phage tRNA-Tyr or host tRNA-Tyr from a plasmid using the phage promoter, suggesting that restoring tRNA levels restores infection. The authors claim that expression of the same tRNAs from a plasmid with the host tRNA promoter has a weak effect. However to my eye the difference is quite modest, my conclusion would be that expression of the tRNA from either promoter provides substantial retron resistance. Are the tRNAs being overexpressed regardless, since they are encoded on plasmids? If conclusions are drawn about tRNA levels and retron resistance, then tRNA levels should be measured under these different rescue conditions.

Answer:

We appreciate your valuable comments and suggestions. We apologize for the oversight in omitting important labeling, such as the statistical significance and the source of the promoter used for tRNA expression, in Extended Data Fig. 6D. We have now revised the figure and quantified phage titers in Extended Data Figs. 6C and 6D, respectively.

In these figures, we demonstrate that tRNAs expressed from plasmids under the host tRNA promoter exhibit a weaker effect compared to those expressed under the phage tRNA promoter in rescuing phage SP15m (which lacks tRNA-Tyr). **Statistical analysis indicates a significant difference ($P < 0.05$), as shown in Extended Data Fig. 6D.** However, we agree with your comment that tRNAs may still be overexpressed in both conditions, as they are encoded on plasmids.

We agree that quantifying tRNA levels is necessary to draw definitive conclusions regarding tRNA levels and retron resistance. Unfortunately, we were unable to directly measure tRNA levels (e.g., via qPCR) due to the challenges posed by the high GC content and the secondary structure of

tRNAs. However, we would like to highlight the findings from Supplementary Fig. S14, which show that red fluorescent protein (RFP) expressed under the phage-derived tRNA promoter exhibits significantly stronger expression compared to RFP expressed under the *E. coli* tRNA-Tyr promoter. Based on this, we believe that the strength of the tRNA promoter may be linked to the critical role of tRNA expression levels in conferring resistance to retron activity.

4. PrrC tests – If I understand correctly, the authors observe resistance to the tRNA-cleaving PrrC system (which protects against T1 and T7 infection) upon expression of a tRNA-Lys from an unrelated phage (SP15). This seems very artificial, and does not provide evidence of a phage that encodes a tRNA to overcome the PrrC system.

Answer:

We understand that the tRNA-Lys used in our study comes from an unrelated phage, which makes the experiment seem artificial. This was due to the limited range of phages we worked with, specifically the T-series phages (T1 to T7). PrrC170 targets T1 and T7, but unfortunately, neither of these phages encodes their own tRNA, making it difficult to use them as models. To address this, we decided to use tRNA-Lys from several phages that belong to different groups. Our results show that phage-derived tRNA-Lys from various sources can help phages overcome the PrrC system (Extended Data Fig 9). Therefore, we believe that in nature, phages may use tRNA as one of their strategies to counteract host defense systems like PrrC. We included this explanation into discussion (Lines 145-147, Lines 203-210).